# Broadband Dynamic Phasor Measurement Method for Harmonic Detection

**Yufu Guo, Hang Xu * and Aobing Chi**

College of Electrical Engineering, Sichuan University, Chengdu 610065, China;
2019141480020@stu.scu.edu.cn (Y.G.); 2019141440225@stu.scu.edu.cn (A.C.)
* Correspondence: xh_scuedu@163.com

**Abstract:** A large number of nonlinear loads and distributed energy sources are connected to the power system, leading to the generation of broadband dynamic signals including inter-harmonics and decaying DC (DDC) components. This causes deterioration of power quality and errors during power measurement. Therefore, effective phasor estimation methods are needed for accurate monitoring and effective analysis of harmonics and interharmonic phasors. For this purpose, an algorithm is proposed in this paper that is implemented in two parts. The first part is based on the least square method in order to obtain accurate DDC component. In the second part, a Taylor–Fourier model of broadband dynamic harmonic phasor is established. The regularization optimization problem of the sparse acquisition model is solved by harmonic vector estimation method. Finally, the piecewise Split-Bregman Iterative (SBI) framework is used to obtain the estimated value of the harmonic phasor measurement and to realize the reconstruction of the original signal. Through simulation and performance test, the proposed algorithm significantly improves the accuracy of the phasor measurement and estimation, and can provide a reliable theoretical basis for the PMU measurement.

**Keywords:** DDC component; broadband dynamic signal; sparse acquisition model; SBI

## 1. Introduction

With the rapid development of smart grids and new energy generation, a large number of nonlinear devices [1] are being applied in the power systems, leading to an increase in harmonics, interharmonics, and DDC components in power grids [2]. On one hand, harmonics and interharmonics have spectral interference with the frequency of interest [3], which affects the measurement accuracy of the electrical energy in the networks [4]. On the other hand, when a fault occurs in the power system, the attenuated DC components of electrical signals cannot be filtered out using traditional algorithms and thus leads to large errors in the calculation results. Therefore, effective phase estimation algorithms [5] are needed to accurately measure the wideband dynamic phase [6].

Currently available phase measurement methods can be divided into two main categories: the discrete Fourier transform (DFT) algorithms [7–9] and non-DFT algorithms. A DFT algorithm is based on equally spaced sampling of the spectrum of a discrete signal in the finite time domain and discretization of the frequency domain function. When the traditional DFT transform algorithm is used to measure the phase, the actual frequency shift results in information redundancy, mutual harmonic interference [8], and information leakage, which cause large measurement errors. Some studies have proposed to introduce over-zero detection method, wavelet transform method, and instantaneous value method [10] to optimize the DFT algorithm. However, these types of methods still have large errors at non-nominal frequencies. The study presented in [11] describes the application of an improved second harmonic filtering technique for single-phase phase measurements at non-nominal frequencies. The method can integrate uniform sampling and fixed-window-length phase measurement with a good performance without changing

the structure of the conventional phase estimation method. However, the DFT-based phase estimation algorithm under the static model offers a poor dynamic performance.

To eliminate the disadvantage that the DTF model is limited to static signal analysis only, some investigations propose to extend the dynamic phase estimation [12–21] to a different mathematical framework, which step up a new generation of phase measurement methods. Such methods apply a specific mathematical model to define the amplitude and phase variations of the signal over a stable sampling interval that is free from the limitations of the period in Fourier analysis.

Currently, dynamic harmonic analysis methods, such as Taylor–Fourier transform (TFT), can achieve phase estimation in the presence of power oscillations, but are susceptible to interference from interharmonics and higher harmonics [12]. In the literature [13], a dynamic harmonic synchronization algorithm based on interpolation function is proposed which broadens the frequency range of TFT and overcomes the limitation of DFT periodicity. However, the method has several complex system parameters and there is a problem of large error after multiple fitting, and the measurement error increases with the increase of the order. The DFT interpolation method with high resolution is proposed in [14], which greatly improves the estimation accuracy. This algorithm has the ability to recover the stage information and suppress the noise by blocking the number of iterations. However, the DFT sequence in this method can only be re-sampled at the nominal rate, which undoubtedly adds uncertainty to the effect.

Based on the above-mentioned methods, the matrix transformation coefficients are obtained by inverse operations of higher-order matrices [15]. Although a number of efficient error reduction algorithms are available, still they cannot address the effects of the superposition of harmonics and noise at higher orders. Thus, machine learning algorithms [16–18] are considered to reduce the computational complexity of the phase solution. The literature [17] limits the effect of interharmonics by identifying the most relevant components of the signal and effectively tracking the harmonics for modeling estimation. However, this methodology is only applicable to low frequency band harmonic vectors. Furthermore, the phase estimation parameters cannot be obtained directly due to the time-varying nature of higher harmonic signals and the high complexity of solving the higher order pseudo-inverse matrices. Considering the field of dynamic harmonic analysis, the work presented in [18] uses the STWLS algorithm for iterative estimation of harmonic parameters and compensates the effect of harmonic parameters on the estimation. This method can improve the frequency measurement accuracy and allocate more space for dynamic information, but it is still difficult to avoid the interference of inter-constrained harmonics. In the literature [19], O-sample dynamic harmonic analysis is proposed, which reduces the computational complexity of DTTFT in closed form and provides the best data compression algorithm for oscillations. However, under non-ideal conditions, the interharmonic interference becomes more and more severe as the order increases due to the large noise in the spatial step reconstruction process.

To solve the synchronous phase estimation solution problem, the study presented in [20] proposes the complex wavelet transform based on the fast Fourier transform (FFT) algorithm to segment the original waveform and perform the signal reconstruction. However, in continuous multiscale analysis, this may lead to severe phase distortion and loss of time information for the next steps of signal processing. In [21], the estimation of frequency, amplitude, and phase in harmonics are obtained by improving the Taylor weighted least squares algorithm. As this method is only applicable to measure third-order continuous waves but not to high-precision harmonic analysis, the algorithm is still not applicable for the measurement of higher-order harmonics.

To address the aforementioned problems, this paper firstly uses Sinc interpolation function to parameterize the attenuated DC component, and uses the least square method to rapidly fit the DDC component in the signal with high precision within the time required by PMU. This method has the advantages of high estimation accuracy and fast response time, which provides reliable research help for the existing DDC component detection. Then, for

broadband harmonic component estimation, an improved Taylor–Fourier model is used which is combined with a machine learning algorithm for accurate recovery of a specific signal with fewer data points. This in turn transforms the harmonic phase reconstruction problem into a sparse acquisition model solving problem. In the regularization-based framework, this paper formulates the phase reconstruction problem as a minimization function. The above underdetermined inverse problem is efficiently solved by using the iterative minimization operation in the SBI system, thereby obtaining the harmonic phase estimation efficiently and accurately. To justify the model, the cross-entropy objective function is used to evaluate the error range of the estimates. The simulation results show that the algorithm has high accuracy and speed in estimating the dynamic phase for wide frequency, and maintains a certain superiority among the compared algorithms. The dynamic performance and anti-interference capability of the algorithm are also validated.

## 2. Estimation of Attenuated DC Component in Broadband Dynamic Signals

### 2.1. Broadband Dynamic Signal Model

For the power system waveform $U(t)$ with DDC component and basic component, the expression is shown in (1):

$$
\begin{aligned}
U(t) &= U_1(t) + U_0(t) \\
&= \sum_{i=0}^{Z} \mathrm{P}_i(t)\cos(2\pi f_i t + \varphi_i(t)) + \lambda(t)e^{-\frac{t}{\tau(t)}}
\end{aligned}
\tag{1}
$$

where $U_1(t)$ is the signal containing fundamental, harmonic, and interharmonic components. $\mathrm{P}_i(t)$ is the time-varying amplitude, $\varphi_i(t)$ is the time-varying phase angle, the reference frequency is $f_i$, and $Z$ refers to the highest harmonic component. $U_0(t)$ is the attenuated DC component signal, $\lambda(t)$ is the attenuated DC component amplitude, and $\tau(t)$ is the time constant.

The dynamic phase model of the signal $U_1(t)$ containing fundamental, harmonic and interharmonic components at a time $t$ is given by:

$$
\mathrm{E}_i(t) \simeq \frac{\mathrm{P}_i(t)}{\sqrt{2}} e^{j\varphi_i(t)}
\tag{2}
$$

Substituting Equation (2) into Equation (1) yields the dynamic expression for the wide-band dynamic signal $U(t)$:

$$
\begin{aligned}
U(t) &= \mathrm{Re}\left[\sqrt{2}\mathrm{E}_i(t)e^{2\pi f_i t}\right] + U_0(t) \\
&= \frac{1}{\sqrt{2}}\left[\mathrm{E}_i(t)e^{2\pi f_i t} + \mathrm{E}_i(t)^* e^{-2\pi f_i t}\right] + U_0(t)
\end{aligned}
\tag{3}
$$

where $*$ is the conjugate operator.

### 2.2. Attenuated DC Component Estimation

The attenuated DC component $U_0(t)$ can be represented by the cosine component sum of the amplitude and phase time variation:

$$
U_0(t) \approx \sum_{n=1}^{Y} \mathrm{P}_{dn}(t)\cos(2\pi f_{dn}t + \varphi_{dn}(t))(-T/2 < t < -T/2)
\tag{4}
$$

where $f_{dn}$, $\mathrm{P}_{dn}(t)$, and $\varphi_{dn}(t)$ denotes the nth cosine component frequency, amplitude, and phase, respectively, while $T$ is the observation interval. In order to meet the accuracy requirements, $Y$ is set to 3 in this study.

Based on the frequency domain sampling theorem, the time domain signal parametric modeling of the decaying DC component is performed using the Sinc interpolation func-

tion, where the dynamic phase of the DDC component is denoted as $x_d = P_d e^{j\varphi_d(t)}/\sqrt{2}$, expressed as:

$$x_d(t) \approx \sum_{n=1}^{3} \sum_{k=0}^{K} x_{k,n} \frac{\sin(\pi f_b t - \pi(k-1))}{\pi f_b t - \pi(k-1)} \tag{5}$$

where $f_b$ is the sampling frequency and $K$ represents the maximum model order. $K$ is set to 2 in this study. $x_{k,n}$ is the sample phase of the nth cosine component of Equation (4) at time $t = k/f_b$.

Sampling the attenuated DC component $U_0(t)$, the discrete form of the fitted DDC component in Equation (4) can be represented by (5):

$$U_0 = \frac{\sqrt{2}}{2}(\Phi_d x_d + \Phi_d^* x_d^*) = \frac{\sqrt{2}}{2}\Theta x_d \tag{6}$$

where $U_{k,n}$ denotes the column vector of the cosine signal for the phase $U_0(t)$. The elements of the matrix $\Phi_d$ are the sample points of $\Phi_{k,n}$ in the sampling window. $\Theta$ denotes the matrix $[\Phi_d \Phi_d^*]$, and $\Phi_{k,n}$ is defined as in (6):

$$\Phi_{k,n} = \frac{\sin(\pi f_b t - \pi(k-1))}{\pi f_b t - \pi(k-1)} e^{j2\pi f_{dn} t} \tag{7}$$
$$(n = 1, 2, 3 \quad k = 0, 1, \ldots, K)$$

The estimated value of the phase $\hat{x}_d$ is found by the least squares method. The calculation procedure is presented in Equation (7). In the matrix $\hat{x}_d$, $\hat{x}_{k,n}$ is the element of this matrix.

$$\hat{x}_d = \sqrt{2}\left(\Theta^H \Theta\right)^{-1} \Theta^H U_0 \tag{8}$$

where $^H$ is the conjugate transpose symbol.

The $\Phi_{k,n}$ of each cosine component is constructed using Equation (8). The matrix $\Theta$ is reconstructed with $\Phi_{k,n}$. The amplitude and phase of each cosine signal is obtained by Equation (5) which leads to the estimation of the DDC component $U_0(t)$.

## 3. Harmonic Phase Estimation in Wideband Dynamic Signals

### 3.1. The Establishment of Harmonic Phasor Estimation Model

In this paper, the authors consider that the Taylor Fourier method is able to describe the amplitude and phase transformations in terms of time during the observation interval. Thus, the $K$ order Taylor expansion model is used to define the dynamic phase quantities as:

$$E_i(t) = \sum_{k=0}^{K} \frac{t^k}{k!} \cdot x_i^{(k)} - \frac{T}{2} < t < \frac{T}{2} \tag{9}$$

where $x^{(k)}$ represents the order derivative of $E(t)$ at $t = 0$, $K$ is the order of Taylor expansion, and $T$ is the duration of the observation interval.

In practical applications, sinusoidal signals containing harmonics and interharmonics are generally in the form of discrete sequences $U_1[n]$ with sequence length $N$ and $-N/2 \le n \le N/2 - 1$. The dynamic phase method is extended to the actual sequence model by multi-frequency phase analysis and the Taylor–Fourier coefficient for each component of the phase of the discrete signal can be expressed as:

$$U_1[n] = \frac{1}{\sqrt{2}}\sum_{i}\left[\sum_{k=0}^{K} \frac{(nH)^k}{k!} x_i^{(k)} e^{j2\pi f_i nH} + \sum_{k=0}^{K} \frac{(nH)^k}{k!} x_i^{*(k)} e^{-j2\pi f_i nH}\right] \tag{10}$$

where $H$ is the sampling interval and the observation interval $T = NH$. Here, $x_i^{(0)}$ is the average harmonic and interharmonic phase and $x_i^{(k)}$ is the kth order derivative of the phase

frequency at $f_i$. $f_i$ is an integer multiple of the fundamental frequency $f_1$ representing the harmonic frequency and a non-integer multiple representing the interharmonic frequency.

The harmonic signal frequency is normalized at a sampling rate of size $1/H$. The sampling length is $N$ to obtain the harmonic component frequency $\delta = f_i H$. The normalized frequency of the Taylor–Fourier basis vector is $\delta_r = r/N$, $r = 0, 1, 2 \cdots, N - 1$. The frequency resolution corresponding to the set of $N$ coefficients is $\Delta = 1/NH$. Considering that the Taylor open order of each $i$ is $K$, Equation (10) can be written in matrix form (11):

$$U_1 = Ax + e \tag{11}$$

where $U_1 = [U_1[-\text{N}/2], \cdots, U_1[\text{N}/2 - 1]]^T$ is the sample column phase. The sensing matrix $A$ of size $N \times [(K + 1)N]$ is the Taylor–Fourier basis phase, $e$ denotes the noise or measurement error, and $x$ is the column phase of length d, describing the set of $x_i = [x_i^{(0)}, \ldots, x_i^{(K)}]$ for the harmonic $i \in \{0, \ldots, N - 1\}$.

As a common practice, the phase solution can be converted to a pseudo-inverse matrix solution using the least squares method. The solution obtained by the pseudo-inverse matrix calculation corresponds to the smallest Euclidean norm, when it is the optimal solution satisfying the constraints described as:

$$\min \|Ax - U_1\|_2 \tag{12}$$

where $\|\cdot\|_2$ denotes the Euclidean parametrization. Based on Lagrangian operator the dynamic phase coefficient estimation matrix $\hat{x}$ is obtained as:

$$\hat{x} = \left(A^H A\right)^{-1} A^H U_1 \tag{13}$$

However, as the maximum harmonic order increases, the matrix dimension increases rapidly, the computational complexity to solve the multivariate linear equation corresponding to Equation (13) increases significantly. For high-frequency harmonics, the least-squares-based phasor method has the lowest computational complexity. Therefore, this phasor method suffers from low efficiency in solving multidimensional matrices. In addition, the conflicting observation time and frequency resolution can lead to less accurate frequency analysis and low time resolution.

*3.2. Reconstructed Harmonic Phase Estimation Model*

Broadband harmonic phasor has certain sparse characteristics [22], which can be transformed into a compressed sensing problem. In order to obtain more accurate results in frequency domain, this paper introduces a correction factor in the harmonic phase analysis $\gamma$. The frequency resolution of each group reaches $\Delta' = \Delta/\gamma$, the length of the sampling sequence is $N' = \gamma \cdot N$, and the sampling length $H$ is 5 fundamental periods. The harmonic component normalized frequencies can be expressed as $\delta'_r \cong r/N'$, $r = 0, 1, 2 \cdots, N' - 1$.

An improved formulation for the Taylor–Fourier coefficients is obtained under more precise correction conditions, expressed as:

$$U_1[\omega/N] = \frac{1}{\sqrt{2}} \sum_i \left\{ \sum_{k=0}^{K} \left[ \frac{1}{k!} \left(\frac{j}{2\pi}\right)^k \left(\frac{\omega}{N} - \delta'_i\right) \right] (\Psi)^{(k)} x_i^{(k)} \right\} \tag{14}$$

where $0 \le \omega < N$, $\Psi$ denote the curvilinear wave transform matrix of size $\Delta N' \times (K + 1)\Delta N'$ and $\Psi^{(k)}$ is the kth order derivative of $\Psi$.

According to Equation (10), the approximate second-order dynamic phase estimate $\hat{\text{E}}_i$, harmonic frequency $\hat{f}_i$ and rate of change of frequency $d\hat{f}/dt$ are obtained by:

$$\hat{\text{E}}_i \cong \sqrt{2}\hat{x}_i^{(0)} \tag{15}$$

$$\hat{f}_i = \Delta'N'\delta'_i + \frac{1}{2\pi} \cdot \mathrm{Im}\left[\hat{x}_i^{(1)}/\hat{x}_i^{(0)}\right] \tag{16}$$

$$\begin{aligned}
\frac{d\hat{f}_i}{dt} &= \frac{1}{2\pi} \cdot \mathrm{Im}\left[\hat{x}_i^{(1)}/\hat{x}_i^{(0)}\right]' \\
&= \frac{1}{2\pi} \cdot \mathrm{Im}\left\{\left[x_i^{(2)}\hat{x}_i^{(0)} - \left(\hat{x}_i^{(1)}\right)^2\right]/\left(\hat{x}_i^{(0)}\right)^2\right\}
\end{aligned} \tag{17}$$

Since the number of harmonic unknowns is much larger than the observed values, it is assumed that $x$ is compressible in the curvelet transform domain [23]. When the correction factor $\gamma \to 0$, and if the sensing matrix $A$ is highly uncorrelated with the sparse matrix $\boldsymbol{\Psi}x$, Equation (14) is transformed into matrix form as:

$$U_1 = \boldsymbol{\Phi}^H\boldsymbol{\Psi}x + e \tag{18}$$

where $U_1$ is the measurement matrix and $x$ denotes the sequence of samples corresponding to each data block of length $\Delta N' \times 1$. $\boldsymbol{\Psi}x$ is the superposition of a small number of phases extracted from the original sample segmentation and $\boldsymbol{\Phi}^H$ is an orthogonal matrix of dimension $N \times N$. Each column of this matrix is the base phase of the DFT where $e$ denotes the noise signal phase.

After the above method realizes the construction of a sparse sampling measurement model, this paper uses the Euclidean search algorithm to matrix chunk $x$ based on the sparsity of the harmonic frequency domain distribution. The $m$ best matching data points are found in the search range, and all data points are combined into a matrix $X_i = \left[x_{i0}, x_{i1}, \cdots x_{i(m-1)}\right]$, $i$ being the coordinates of the coefficient matrix. Curvilinear transformation is performed on the matrix, and the insignificant curvilinear coefficients are removed using the curvilinear threshold criterion, and the extracted coefficients require effective denoising and compression algorithms. The curvilinear transform is performed, the insignificant curvilinear coefficients are removed using the curvilinear threshold criterion, and the extracted coefficients require an effective denoising and compression algorithm.

To ensure the local smoothness feature of harmonic detection, the sparse matrix $\boldsymbol{\Psi}x$ contains only a small set of non-zero valid terms. The phase quantity of the transform coefficient column obtained by arranging this set of valid terms in the dictionary order is $\boldsymbol{\Omega}_x$. As the transform coefficients $\boldsymbol{\Omega}_x$ are concentrated around the zero region and distributed in the form of fine peaks, the distribution characteristics of the transform coefficients are used to characterize the repeatability of the signal and are denoted as:

$$\boldsymbol{\psi}(x) = \|\boldsymbol{\Omega}_x\|_{l1} \tag{19}$$

where $\|\cdot\|_{l1}$ denotes the parametric number $l1$. A more accurate estimation of the harmonic signal for each group can be achieved by inverting the transform coefficients as:

$$\widetilde{x} = \boldsymbol{\psi}^{-1}(\boldsymbol{\Omega}_x) \tag{20}$$

where $\boldsymbol{\psi}(x)^{-1}$ is the inverse operator of $\boldsymbol{\psi}(x)$.

### 3.3. Solution of Harmonic Estimation Model

In order to obtain a suitable estimation accuracy and robustness against noise, firstly the harmonic phase recovery model is established based on the above presented method. This leads to the reconstruction of the Taylor coefficient matrix $x$ by solving the optimization problem of the regularized linear least squares cost function, using:

$$\min_x\left\{\frac{1}{2}\left\|U_1 - \boldsymbol{\Phi}^H\boldsymbol{\Psi}x\right\|_{l2}^2 + \sigma R(x)\right\} \tag{21}$$

where $\|\cdot\|_{l2}$ denotes the parameter $l2$. $R(x)$ denotes the regularization term $R(x) = \rho\|\mathbf{\Psi}x\|_{l1} + \|\mathbf{\Omega}_x\|_{l1}$ and $\sigma$ is the regularization factor. $\|\mathbf{\Psi}x\|_{l1}$, $\|\mathbf{\Omega}_x\|_{l1}$ denote the local and non-local sparse terms, respectively. $\rho$ is a regularization parameter to balance the two sparse terms.

To solve the above minimization problem, an alternating SBI algorithm is used in this paper [24]. Two auxiliary phase quantities $p$ and $q$ are introduced, and the following scheme is finally obtained:

$$x^{k+1} = \arg\min \frac{1}{2}\left\|U_1 - \mathbf{\Phi}^H\mathbf{\Psi}x\right\|_{l2}^2 + \frac{\mu_1}{2}\left\|(\mathbf{\Psi}x)^k - \mathbf{\Psi}x - p^k\right\|_{l2} + \frac{\mu_2}{2}\left\|x - \widetilde{x}^k - q^k\right\|_{l2}^2 \quad (22)$$

$$(\mathbf{\Psi}x)^{k+1} = \arg\min \rho'\|\mathbf{\Psi}x\|_{l1} + \frac{\mu_1}{2}\left\|\mathbf{\Psi}x - \mathbf{\Psi}x^{k+1} - p^k\right\|_{l2}^2 \quad (23)$$

$$\widetilde{x}^{k+1} = \arg\min \sigma\|\mathbf{\Omega}_x\|_{l1} + \frac{\mu_2}{2}\left\|x^{k+1} - \widetilde{x} - q^k\right\|_{l2}^2 \quad (24)$$

$$p^{k+1} = p^k - (\mathbf{\Psi}x)^{k+1} + \mathbf{\Psi}x^{k+1} \quad (25)$$

$$q^{k+1} = q^k - x^{k+1} + \widehat{x}^{k+1} \quad (26)$$

where $\mu_1$ and $\mu_2$ are constants that serve to improve the stability of the algorithm values. $\rho' = \sigma\rho$. $p, q$ are the SBI algorithm auxiliary iterative phase quantities.

The minimization of a strictly convex quadratic function is described by Equation (22). The corresponding closed solution can be obtained by setting the gradient of the objective function to 0:

$$x^{k+1} = \left[\mathbf{\Psi}^T\left(\mathbf{\Phi}^H\right)^T\mathbf{\Phi}^H\mathbf{\Psi} + (\mu_1 + \mu_2)I_n\right]^{-1} \cdot \left[\mu_1\mathbf{\Psi}^T\left((\mathbf{\Psi}x)^k - p^k\right) + \mu_2\left(\widetilde{x}^k + q^k\right) + \mathbf{\Psi}^T\left(\mathbf{\Phi}^H\right)^T U\right] \quad (27)$$

where $I_n$ is a unit matrix of size $n \times n$. The problem of minimization of the function can be solved using the step-optimal, which is considered as a faster method [25], and can be expressed as:

$$x^{k+1} = x^k - d^k g^k \quad (28)$$

where $g$ is the gradient of the objective function and $d$ is the optimal step size expressed as:

$$d = abs\left(g^T g / g^T\left(\mathbf{\Psi}^T\left(\mathbf{\Phi}^H\right)^T\mathbf{\Phi}^H\mathbf{\Psi} + (\mu_1 + \mu_2)I_n\right)g\right).$$

Considering the estimated value $\mathbf{\Psi}\widetilde{x} = \mathbf{\Psi}x + p$, the problem of (23) can be solved by applying the minimum mean square error theorem as:

$$\min \frac{1}{2}\|\mathbf{\Psi}x - \mathbf{\Psi}\widetilde{x}\|_{l2}^2 + \frac{\rho'}{\mu_1}\|\mathbf{\Psi}x\|_{l1} \quad (29)$$

Similarly, the problem of (24) can be written as:

$$\min \frac{1}{2}\|\widetilde{x} - \boldsymbol{\theta}\|_{l2}^2 + \frac{\sigma}{\mu_2}\|\mathbf{\Omega}_x\|_{l1} \quad (30)$$

Here, $\boldsymbol{\theta} = x - q$ (omitting $k$). Due to the cumbersome definition of $\mathbf{\Omega}_x$, it is difficult to solve the above equation intuitively. In this paper, a set of closed solutions is thus obtained by making reasonable assumptions. Treating $\boldsymbol{\theta}$ as some type of observation $\widetilde{x}$ with noise and denoting the error phase by $e = \widetilde{x} - \boldsymbol{\theta}$, the error of each element is $e(i)$ ($i = 1, 2, 3 \cdots n$), respectively. Further, assuming that $e$ is independently distributed over $(i, i, d)$ with mean 0 and variance $v^2$, it follows from the law of large numbers that for any $\varepsilon > 0$, there are:

$$\lim_{n\to\infty} P\left\{\left|\frac{1}{n}\sum_{i=1}^{n}e_i^2(i) - v^2\right| < \frac{\varepsilon}{2}\right\} = 1 \quad (31)$$

This leads to the expression:

$$\lim_{n \to \infty} P\left\{ \left| \frac{1}{n}\|\widetilde{x} - \theta\|_{l2}^2 - v^2 \right| < \frac{\varepsilon}{2} \right\} = 1 \tag{32}$$

The transformed error phase is $\Omega_e = \Omega_x - \Omega_\tau$, $\Omega_e(j)$ $(j = 1, 2 \cdots m)$ denotes the error phase of each element, and $m$ is the number of data points for the best match. According to the orthogonal nature of the matrix, the transformation does not change the variance of each group. It follows that $\Omega_e$ of each group is independently distributed over $(i, i, d)$, (with zero mean and variance $v^2$), and the large number theorem yields that for any $\varepsilon > 0$, the obtained expression is:

$$\lim_{m \to \infty} P\left\{ \left| \frac{1}{m}\sum_{i=1}^{m} \Omega_e^2(j) - v^2 \right| < \frac{\varepsilon}{2} \right\} = 1 \tag{33}$$

This results in:

$$\lim_{\substack{n \to \infty \\ m \to \infty}} P\left\{ \left| \frac{1}{n}\|\widetilde{x} - \theta\|_{l2}^2 - \frac{1}{m}\|\Omega_x - \Omega_\theta\|_{l2}^2 \right| < \varepsilon \right\} = 1 \tag{34}$$

Therefore, it can be concluded that:

$$\frac{1}{n}\|\widetilde{x} - \theta\|_{l2}^2 = \frac{1}{m}\|\Omega_x - \Omega_\theta\|_{l2}^2 \tag{35}$$

Combining it with the problem (24) results in:

$$\underset{\Omega_z}{\operatorname{argmin}} \frac{1}{2}\|\Omega_x - \Omega_\theta\|_{l2}^2 + \frac{m\sigma}{\mu_2 n}\|\Omega_x\|_{l1} \tag{36}$$

Since the unknown quantity $\Omega_x$ is separable in the above equation, each component can be reduced to:

$$\Omega_x(j) = \underset{\Omega_x(j)}{\operatorname{argmin}}\left\{ \frac{1}{2}|\Omega_x(j) - \Omega_\theta(j)|^2 + \frac{\sigma m}{\mu_2 n}|\Omega_x(j)| \right\} \tag{37}$$

where $|\cdot|$ is the mode corresponding to the phase quantity.

The expression for the estimated value of $x$ in vector contraction form can be written as:

$$\widetilde{x} = \psi^{-1}[\Omega_\theta(|\Omega_\theta| - \sigma m)/(|\Omega_\theta|\mu_2 n)] \tag{38}$$

Estimating the Taylor coefficient matrix $x$ (described above), the reorganization of the harmonic signal $U_1[n]$ is achieved through the harmonic sampling equation $U_1 \simeq \Phi^H \Psi \widetilde{x}$ for accurate detection of the harmonic phase changes. To ensure the accuracy of the model, the cross-entropy [26] objective function is used to measure the difference in probability distribution between the estimated value $\widetilde{x}$ and the theoretical value $x$ in the framework of logistic regression, expressed as:

$$\begin{aligned} L(\widetilde{x}, x) &= -\frac{1}{N}\sum_i x_i \log \widetilde{x}_i - (1 - x_i) \log(1 - \widetilde{x}_i) \\ &= -\frac{1}{N}\sum x \log(\psi^{-1}(\Omega_x)) - (1 - x) \log(1 - \psi^{-1}(\Omega_x)) \end{aligned} \tag{39}$$

Assuming that the error is binary distributed, $L \to 0$ enables the predicted probability distribution to become closely correlated with the actual probability distribution, which proves that the assumptions are consistent with the expected model.

In summary, the specific steps of the algorithm harmonic phase estimation implemented in this paper are shown in Scheme 1.

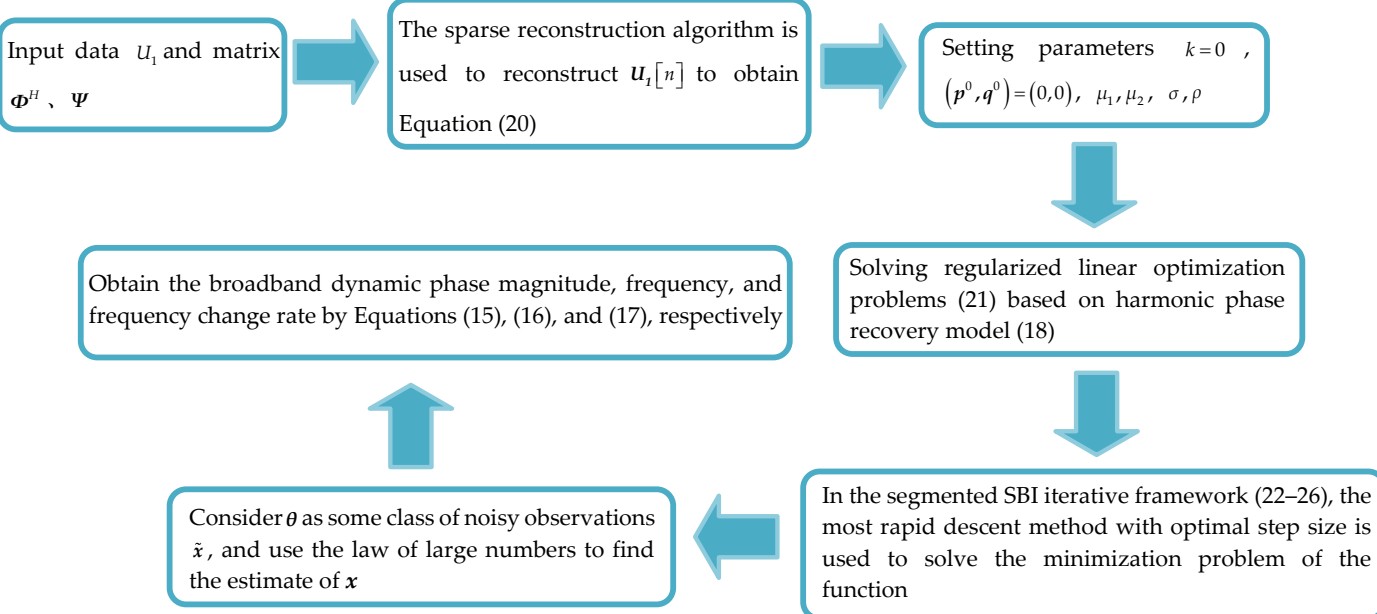

**Scheme 1.** Algorithm flow.

## 4. Simulation Analysis

In this section, the results of the proposed wideband dynamic phasor measurement algorithm (BMP) proposed are discussed considering different test scenarios. Four algorithms compared in this paper are: FFT algorithm [27], Prony algorithm [28], Taylor weighted least squares (TWLS) algorithm [21], and Sinc interpolation function based dynamic simultaneous phase measurement (SIFE) algorithm [13]. Test scenarios include basic performance test, frequency ramp test, step transform test, and interference test.

To compare the estimation effects of different methods, the total phase error (TVE) is determined to describe the relative deviation between the theoretical phase and the estimated phase. TVE is closely related to the amplitude error and phase angle error, but cannot reflect the variation of a single aspect alone. Therefore, this paper also suggests two metrics: frequency error (FE) and absolute rate of change of frequency error (RFE) to comprehensively evaluate the effect of phase estimation. Measurement requirements for M class PMU [29] are selected as the limits for different operating conditions according to the IEEEC 37.118.1 standard.

All five algorithms use the same rectangular observation window with a fundamental frequency bandwidth of 1 Hz. The sampling window length is set to 5 working frequency cycles. Taking into account the complexity of the algorithm and the accuracy of the algorithm, the value of $\gamma$ is set to 20, to improve the frequency domain resolution.

### 4.1. Basic Performance Test

In order to test the basic performance of the proposed algorithm, the wideband dynamic signal model with DDC components is constructed by using:

$$U(t) = \mathrm{Re}\left[ e^{j(2\pi f_0 t + \varphi_0(t))} + 0.1 e^{j(2\pi f_h t + \varphi_h(t))} \right] + \lambda(t) e^{\left( -\frac{t}{\tau(t)} \right)} \tag{40}$$

where $f_0$ is the fundamental frequency which is set to 50 Hz, $\varphi_0(t)$ and $\varphi_h(t)$ denote the fundamental and each harmonic phase angle, respectively. This value of frequency is chosen arbitrarily in the $(-\pi, \pi)$ range. The number of harmonics $h$ in the low frequency band is taken as 2–13 and in the high frequency band $h$ is taken as 77, 79, 80, and 83, respectively. The sampling frequency is set to 10 kHz. The values of the DDC component amplitude $\lambda(t)$ are taken as 0.3, 0.4, 0.5, . . . 1. The time constant $\tau(t)$ of the DDC component starts from 0.01 s and varies in steps of 0.01 s to 0.1 s.

When applying the BMP algorithm, iterating the initial value of the phase $(p^0, q^0) = (0, 0)$, the regularization factor is considered as $\sigma = 2.7$. For the fixed-value parameters $\mu_1 = \zeta\mu$, $\mu_2 = (1 - \zeta)\mu$, $\mu = 2.7 \times 10^{-3}$, $\zeta$ can vary in the range [0.05, 0.3]. The number of data points that are best matched in the search window is set to $m$. In the process of matrix chunking, if the number of matching blocks is too large, there must be data points in the block array with high noise impact and low matching. Conversely, the influence of chance on the construction matrix is unavoidable. In this paper, the maximum number of iterations is J. The higher the number of iterations, the higher the computational accuracy. However, the computational cost becomes significantly higher. This paper combines the analysis of reconstruction effect and algorithm running time by changing the parameters m and J. The corresponding results are shown in Figure 1.

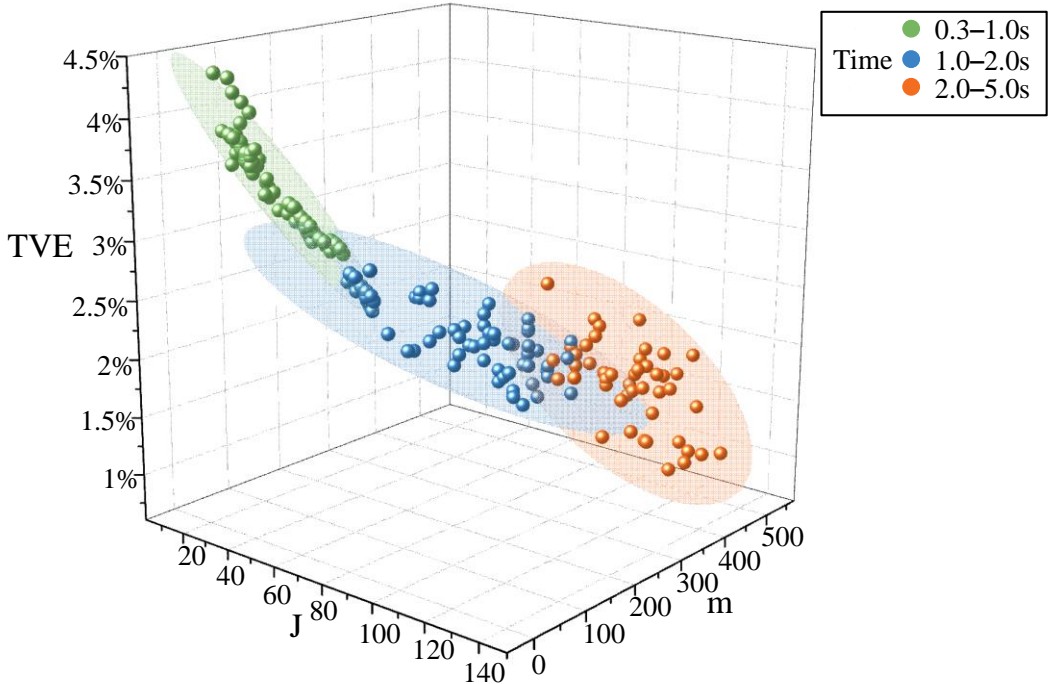

**Figure 1.** Reconfiguration effect and running time range with parameters.

As can be seen from Figure 1, the total phase error tends to decrease as the number of parameters (*m* and J) increases, but the running time algorithm increases as well. When the number of data points *m* = 240 and the number of iterations J is less than 100, changing the number of iterations has a minor impact on the computational cost (small change in running time). In this case, the reconstruction effect tends to be stable. When the number of iterations increases, the computational complexity increases significantly (large change in running time) and the TVE shows an unstable decreasing trend. Therefore, an appropriate reduction of m can effectively improve the reconstruction effect. Considering both reconstruction performance and running time, the parameters *m* = 240 and the number of iterations J = 100 are used in the algorithm.

FFT, Prony, TWLS, and SIFE are chosen as the comparison algorithms, and the results of the total phase error estimation, frequency and frequency change rate error estimation are shown in Table 1.

**Table 1.** Comparison of estimation accuracy among algorithms.

| Index | h | BMP | Prony | FFT | TWLS | SIFE | Index | h | BMP | Prony | FFT | TWLS | SIFE | Index | h | BMP | Prony | FFT | TWLS | SIFE |
|---|---|---|---|---|---|---|---|---|---|---|---|---|---|---|---|---|---|---|---|---|
| TVE (%) | 1 | 2.79 | 5.79 | 8.66 | 8.29 | 2.55 | FE (Hz) | 1 | 0.092 | 0.78 | 1.88 | 1.93 | 0.97 | RFE (Hz/s) | 1 | 2.437 | 6.73 | 5.13 | 6.12 | 4.85 |
| | 2 | 2.67 | 4.45 | 8.62 | 5.41 | 2.25 | | 2 | 0.084 | 0.76 | 1.23 | 1.32 | 0.99 | | 2 | 2.376 | 5.52 | 4.96 | 5.96 | 2.36 |
| | 3 | 2.7 | 3.3 | 6.02 | 5.31 | 2.3 | | 3 | 0.096 | 0.7 | 1.16 | 1.24 | 0.87 | | 3 | 2.403 | 4.2 | 4.76 | 5.8 | 1.23 |
| | 4 | 2.61 | 2.65 | 4.37 | 5.27 | 2.1 | | 4 | 0.076 | 0.66 | 0.75 | 0.98 | 0.72 | | 4 | 1.377 | 3.8 | 4.4 | 5.56 | 0.957 |
| | 5 | 2.49 | 2.2 | 3.36 | 5.21 | 2.25 | | 5 | 0.064 | 0.7 | 0.74 | 0.96 | 0.71 | | 5 | 1.161 | 3.88 | 4.38 | 5.5 | 0.845 |
| | 6 | 2.37 | 1.9 | 5.03 | 4.37 | 1.9 | | 6 | 0.047 | 0.62 | 0.81 | 0.95 | 0.65 | | 6 | 0.972 | 3.44 | 4.12 | 5.18 | 0.613 |
| | 7 | 2.19 | 1.7 | 5.99 | 4.21 | 2.35 | | 7 | 0.048 | 0.58 | 0.74 | 0.91 | 0.62 | | 7 | 0.837 | 2.92 | 3.86 | 4.86 | 0.576 |
| | 8 | 2.13 | 1.5 | 3.1 | 3.12 | 2.1 | | 8 | 0.048 | 0.52 | 0.71 | 0.87 | 0.54 | | 8 | 0.729 | 2.68 | 3.94 | 4.74 | 0.491 |
| | 9 | 2.16 | 1.35 | 3.12 | 3.65 | 2 | | 9 | 0.036 | 0.54 | 0.68 | 0.81 | 0.51 | | 9 | 0.648 | 2.48 | 3.72 | 4.38 | 0.369 |
| | 10 | 2.07 | 1.25 | 2.06 | 3.12 | 2.05 | | 10 | 0.04 | 0.52 | 0.67 | 0.813 | 0.61 | | 10 | 0.543 | 2.36 | 3.28 | 4.06 | 0.341 |
| | 11 | 1.86 | 1.15 | 1.75 | 3.07 | 2.45 | | 11 | 0.028 | 0.5 | 0.64 | 0.812 | 0.52 | | 11 | 0.513 | 1.84 | 3.18 | 3.98 | 0.312 |
| | 12 | 1.62 | 1.05 | 1.68 | 2.78 | 1.9 | | 12 | 0.016 | 0.46 | 0.57 | 0.765 | 0.47 | | 12 | 0.459 | 1.76 | 3.06 | 3.78 | 0.311 |
| | 13 | 0.21 | 0.5 | 0.91 | 1.23 | 1.7 | | 13 | 0.013 | 0.38 | 0.39 | 0.58 | 0.75 | | 13 | 0.517 | 1.4 | 2.52 | 1.9 | 0.307 |
| | 56 | 0.18 | 0.55 | 0.87 | 1.09 | 0.16 | | 56 | 0.004 | 0.36 | 0.4 | 0.54 | 0.03 | | 56 | 0.405 | 1.4 | 2.46 | 1.8 | 0.318 |
| | 71 | 0.09 | 0.45 | 0.8 | 0.99 | 0.11 | | 71 | 0.008 | 0.3 | 0.37 | 0.5 | 0.07 | | 71 | 0.351 | 1.24 | 1.52 | 1.58 | 0.313 |
| | 74 | 0.27 | 0.25 | 0.75 | 1.01 | 0.15 | | 74 | 0.004 | 0.22 | 0.34 | 0.4 | 0.14 | | 74 | 0.406 | 1.16 | 1.18 | 1.46 | 0.32 |
| | 82 | 0.09 | 0.4 | 0.64 | 0.93 | 0.07 | | 82 | 0.009 | 0.22 | 0.26 | 0.44 | 0.09 | | 82 | 0.275 | 1.12 | 0.94 | 1.56 | 0.311 |

As can be seen in Table 1, the maximum values of TVE, FE and RFE indices of the BMP algorithm are 2.79%, 0.096 Hz, and 2.437 Hz/s, respectively. When harmonic components are included, the IEEE standard limits for TVE and FE are determined as 3% and 0.1 Hz, respectively, and the limit for RFE is reported as 2.7 Hz/s [28]. Compared with other comparative algorithms, the estimation metrics of this algorithm fully satisfies the requirements of the IEEE standard. The results show that the algorithm still provides a good detection considering the condition of broadband harmonics containing DDC components with the highest accuracy of phase estimation. The BMP algorithm uses the sparse distribution of the harmonic frequency domain distribution to identify the most relevant components of the signal. This improves the accuracy of the measurement results significantly.

The estimation error of the SIFE method near the lower harmonics does not meet the IEEE measurement standard. This is because the SIFE method is based on a low-pass filter for baseband signal filtering. Therefore, it is difficult to obtain zero-error results. However, the algorithm performs better than FFT and TWLS because of its wide passband and wide stopband, which can efficiently estimate the harmonic simultaneous phase. The FFT measurement results are most affected by spectral leakage which will reduce the accuracy of harmonic parameter identification significantly. The maximum total phase error exceeds 8% and the accuracy of its FE and RFE measurements is also unsatisfactory. Under dynamic conditions, the Fourier transform model is not able to track the phase changes in the observation window, resulting in incorrect phase evaluation. The TWLS method uses a second-order Taylor order to fit the signal components. However, the Taylor signal model has large errors and limited accuracy. Increasing the Taylor model order can reduce the model error. The drawback of this approach is that the higher the order, the worse the passband performance of the filter. The Prony algorithm uses a parametric model to calculate the signal parameters. However, its estimation order limits the number of estimated frequency components and the frequency estimation error gradually increases.

*4.2. Frequency Ramp Test*

The power imbalance between the load and the generator causes a decrease in the frequency of the wideband signal as the load increases while it increases as the input power increases. To analyze the performance of the BMP algorithm considering frequency ramping, the provided signals can be expressed as:

$$U(t) = \text{Re}\left[ e^{j(2\pi ft + \pi R_1 t^2 + \varphi_1(t))} + 0.1\sum_h e^{j(2\pi hft + \pi hR_1 t^2 + \varphi_h(t))} \right] + \lambda(t)e^{\left(-\frac{t}{\tau}\right)} \tag{41}$$

where $f$ is the fundamental frequency and takes the value of 50 Hz. $R_1$ is the fundamental frequency slope and takes the value of 1 Hz/s in this paper. $\varphi_1(t)$ and $\varphi_h(t)$ are the fundamental phase and harmonic phase, respectively, and the phase is set as a random number uniformly distributed in the $(-\pi, \pi)$ range.

The results of harmonic phase estimation, frequency estimation, and rate of change of frequency estimation for this paper and the comparison algorithm are shown in Figure 1. It is assumed that the sampling frequency is 10 kHz and the sampling window length is set to 5 work frequency cycles. The FFT, Prony, TWLS, and SIFE are used as comparison algorithms to analyze the signal of Equation (41). The estimation results generated by each method considering the frequency ramp condition are shown in Figure 2.

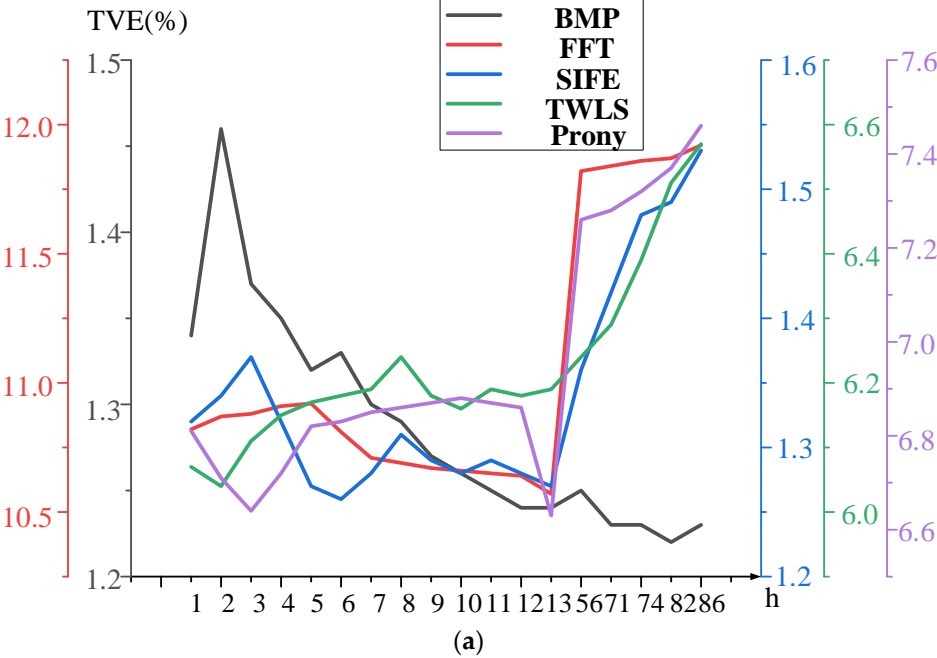

**Figure 2.** *Cont.*

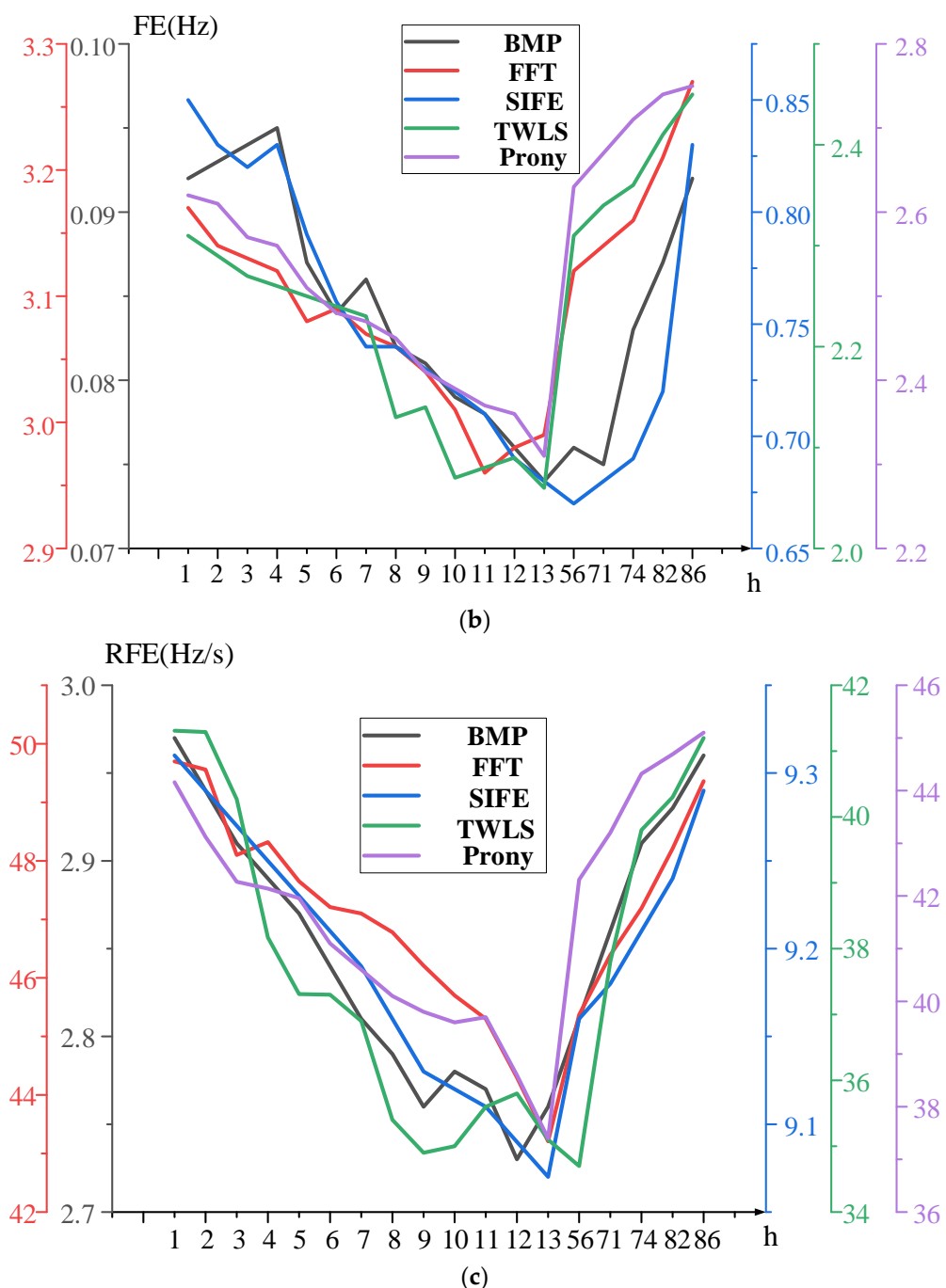

**Figure 2.** Maximum errors of TVE, FE, and RFE under frequency ramp conditions. (**a**) TVE. (**b**) FE. (**c**) RFE.

In Figure 2, it can be seen that the maximum error of TVE algorithm is 1.46%, the maximum frequency error is 0.095 Hz, and the maximum error of frequency conversion rate is 2.97 Hz/s. The obtained error values meet the requirements of IEEE standard [28]. The results show that the proposed TEV algorithm can maintain high accuracy even when the fundamental frequency varies widely and linearly. Its estimation accuracy is better than the other four algorithms. The BMP algorithm uses Taylor's order to approximate the dynamic signal model which not only can estimate the frequency and phase angle accurately, but also is least affected by the linear frequency variation. Therefore, it can be stated that the BMP algorithm has the capability to achieve the highest accuracy of phase estimation. Among the remaining algorithms, the FFT method is not able to track frequency changes in real time

under dynamic conditions, and therefore has a large frequency variation rate. It is observed that the error calculation results of both Prony and TWLS algorithms are smaller than those of FFT. The reason is that the TWLS is a dynamic model-based estimation algorithm and Prony algorithm can accurately extract the low-frequency oscillation eigenvalues of the dominant mode. Furthermore, the error characteristics of parameters such as phase angle and frequency of the algorithm are less affected by the frequency change. However, the measurement results are not able to meet the requirements of IEEE standard to an acceptable level. If enough information about the time-varying characteristics of the signal is unknown, obtaining accurate mode parameters is challenging by using Prony algorithm. Similarly, it is found that TWLS algorithm is also unable to estimate the higher-order harmonic phase quantities accurately due to the influence of the fundamental components.

### 4.3. Step-Transformation Test

In order to simulate fault conditions with sudden changes in the amplitude and phase of voltage/current signals, it is necessary to simulate the proposed algorithm considering these conditions in order to evaluate the response time and delay. At the beginning of the test, the amplitude of each component is set to 115% of the initial amplitude while the phase changes to $(\pi/6)$. The broadband dynamic signals can be expressed as:

$$U(t) = \text{Re}\left[ e^{j(2\pi f_0 t + \varphi_0(t))} + 0.1\sum_h e^{j(2\pi f_h t + \varphi_h(t))} \right] + \lambda(t)e^{\left(-\frac{t}{\tau(t)}\right)} \tag{42}$$

The tests in this section assume a sampling rate of 5 kHz and a sample period length of 5 cycles. In the standard, the speed of response time is used to evaluate the performance of each algorithm under step-change conditions. It is defined as the time interval between the first and the last instant greater than a given threshold. According to the IEEE standard for the test conditions observed in this section, the thresholds for the maximum TVE, FE, and RFE values are 1.5%, 0.13 Hz, and 0.78 Hz/s, respectively. The obtained simulation results are presented in Figure 3.

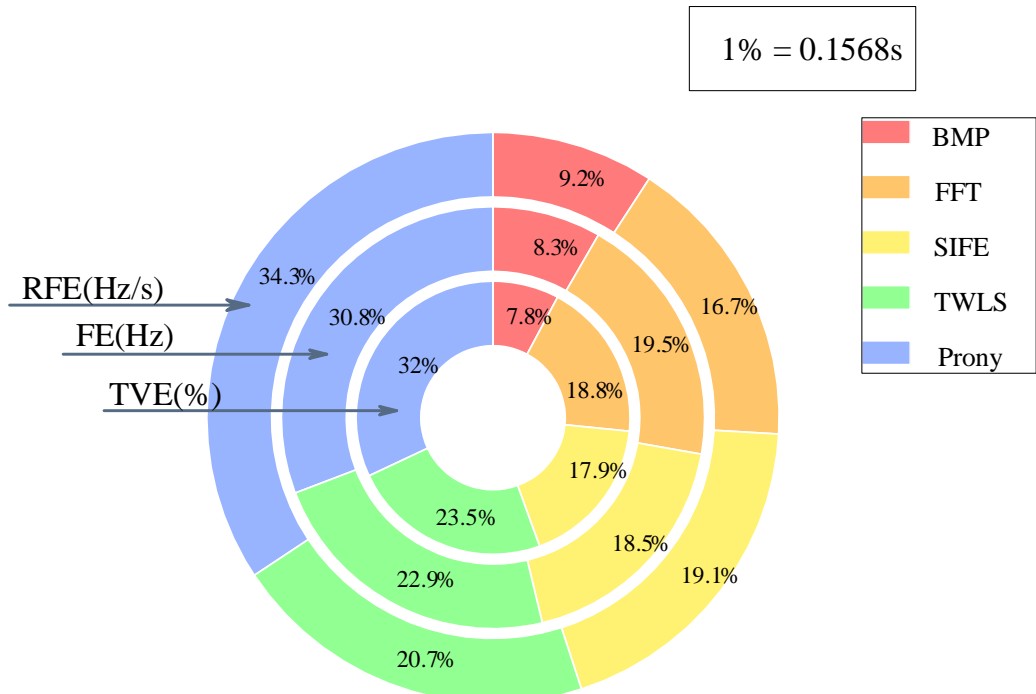

**Figure 3.** Runtime ratio of each method under the step.

As can be seen in Figure 3, the BMP algorithm takes less time to reach the IEEE standard based TVE, FE, and RFE values under the amplitude and phase step-transformation conditions as compared to other algorithms. It can be concluded that the BMP algorithm has the highest estimation accuracy and the response time fully satisfies the standard requirements for P-class PMU under the step-transform condition. For the estimation of the entire Fourier transform of the stepwise smooth function, the FFT algorithm requires a complex multiplication operation. The performance of TWLS algorithm can be improved by recalculating its coefficients in each reporting frame. However, this approach significantly increases the computational burden and requires the calculation of pseudo-inverse. In order to implement the SIFE algorithm to meet the accuracy requirements, a complex matrix with 70 rows and 1400 columns needs to be stored in memory. Considering the processing practicalities, $2^{16}$ real multiplications and $2^{15}$ real additions need to be carried out in real time, which is very limited in terms of memory and processing power. In order to get more accurate results for the Prony algorithm, the model order of the algorithm needs to be increased, thus increasing the computational effort. In this paper, the BMP algorithm uses a machine learning algorithm to collect the key information of the signal and construct the sparse matrix. This solution greatly reduces the complexity of the operation and results in faster and accurate detection of broadband dynamic signals.

### 4.4. Anti-Jamming Test

Typically, the signals flowing in the power grid contains a certain amount of interharmonics and noise, which can seriously affect the estimation of harmonic phase. In this section, Gaussian white noise is introduced with a signal-to-noise ratio of 55 dB. The broadband dynamic signals can be expressed as:

$$U(t) = \text{Re}\left[ e^{j(2\pi f_0 t + \varphi_0(t))} + 0.1 \sum_h e^{j(2\pi f_h t + \varphi_h(t))} \right] + \text{Re}\left[ 0.05 e^{2\pi j f_t t} \right] + \lambda(t) e^{\left( -\frac{t}{\tau(t)} \right)} + \text{noise} \tag{43}$$

where $f_t$ is the interharmonic frequency and its values are considered in the order of 9652.5 Hz, 9751.5 Hz, 9850.5 Hz, 9949.5 Hz, 10,048.5 Hz, and 10,147.5 Hz. Where $\phi_1$ and $\phi_h$ are the phases of the fundamental and harmonics and their values are random numbers in the range of $(0, 2\pi)$. The sampling frequency is assumed as 10 kHz and the sampling window length is set to 5 I.F. periods. The obtained simulation results are shown in Figure 4.

It can be observed in Figure 4 that the TVEmax of the proposed algorithm is 2.43%, FEmax is 0.071 Hz, and RFEmax is 0.184 Hz/s. The calculation based results of the algorithm are found better than the rest of the comparison algorithms. In the presence of interharmonic interference, the IEEE standard limits for TVE, FE and RFE are 3.5%, 0.2 Hz, and 3 Hz/s, respectively. Similarly, the results of TVEmax, FEmax, and RFEmax parameters for SIFE algorithm are 6.64%, 0.115 Hz and 7.39 Hz/s, respectively, while those for Prony algorithm are 19.73%, 0.136 Hz, and 9.207 Hz/s, respectively. The calculation based results of each index of Prony algorithm are better than the other comparative algorithms. The indexes of BMP algorithm meet the requirements of IEEE standard. In case of noise interference in this segment, serious interference and spectral leakage occurs between adjacent harmonics of the FFT algorithm, which affects the resolution and accuracy. Similarly, in case of noise interference in this segment, the TWLS and SIFE algorithm increase the amplitude of the transition band for their harmonic filters, causing interference between adjacent harmonics which leads to large estimation errors. In addition, the TWLS algorithm is severely affected by interharmonic interference, making it challenging to estimate each high-frequency component accurately. For higher-order components, the model parameters of the Prony algorithm are constantly modified as the harmonic order increases. Therefore, Prony algorithm has good results for estimation of interharmonic phase and frequency. This can suppress the effect of spectral leakage of interharmonic components to some extent. However, this method requires pre-estimation of the order of the dynamic time-varying signal.

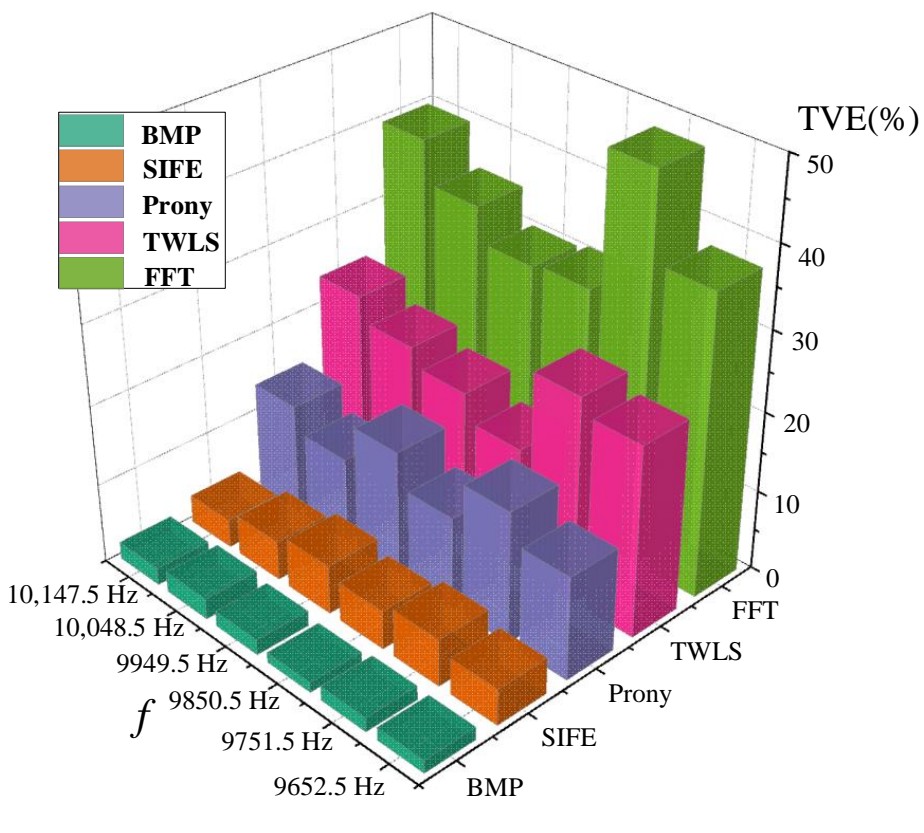

(**a**)

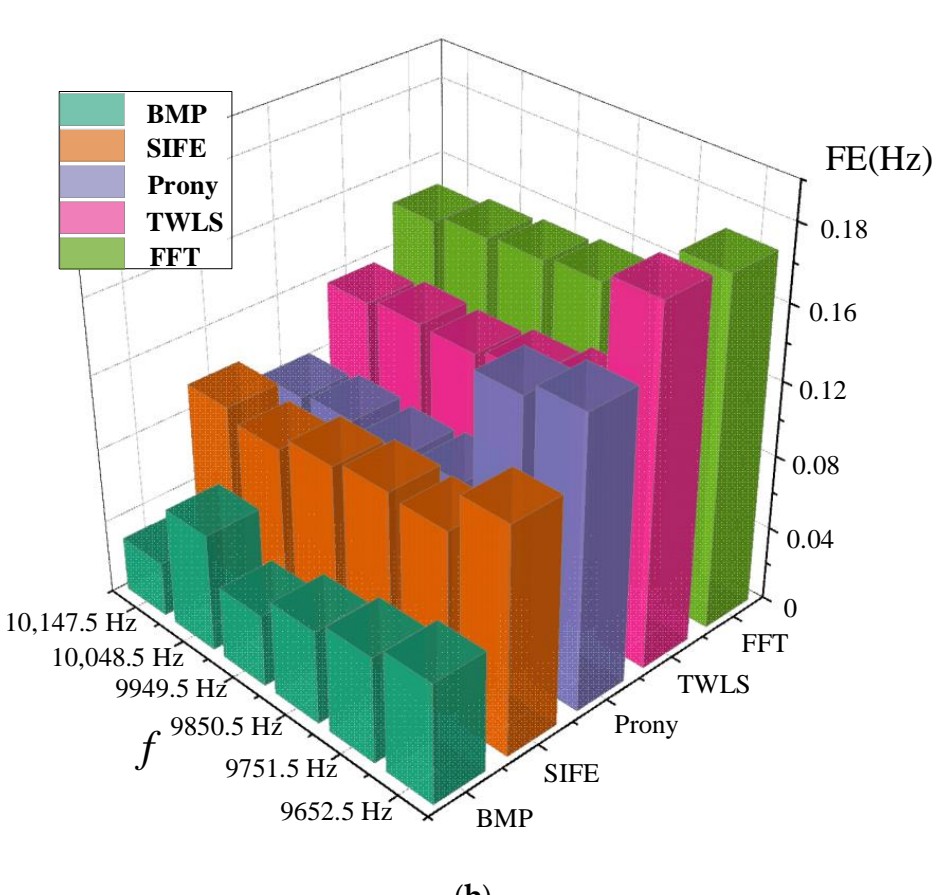

(**b**)

**Figure 4.** *Cont.*

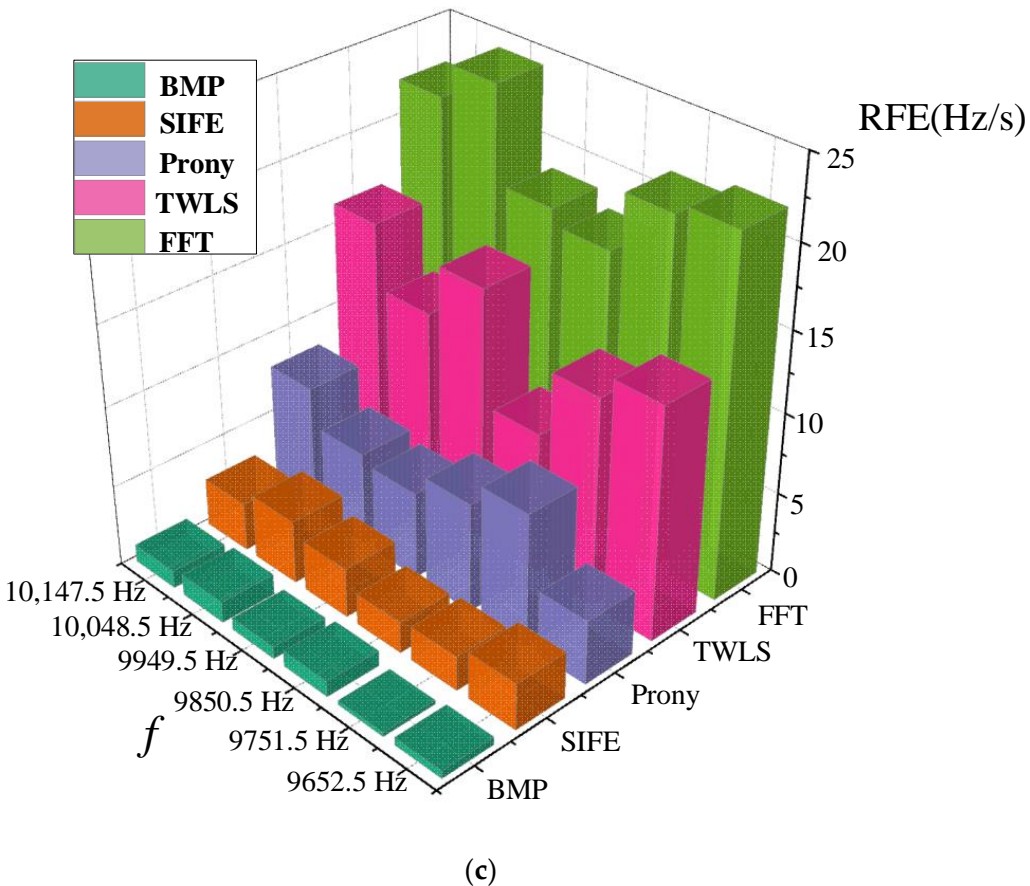

(**c**)

**Figure 4.** Maximum error of TVE, FE, and RFE under interharmonic and noise interference. (**a**) TVE. (**b**) FE. (**c**) RFE.

The BMP algorithm uses a modified Taylor–Fourier model to recover a specific signal with fewer data points accurately. This leads to accurate reconstruction of the harmonic phase to solve with a sparse acquisition model, which effectively improves the reconstruction performance and noise immunity of the algorithm. In addition, the algorithm reconstructs the spectrum with a resolution of 1 Hz, which facilitates the accurate detection of interharmonic components.

*4.5. Measurement Validation and Analysis*

In order to demonstrate the BMP's practical values, we use the current field data recorded at a high-speed rail converter station for testing. The measuring instrument is MHD-AE301 multi-functional power monitoring instrument. In Figure 5, the field data and its spectrum are shown. As seen, there are significant fundamental and third-harmonic components.

We investigate the BMP algorithm in this paper by sampling data in matlab simulation platform. The parameter estimates for the BMP method are shown in Figure 6.

It can be observed in Figure 6 that the amplitude, frequency, or ROCOF estimates for the BMP method are almost identical to those for the power monitor. Therefore, BMP method can be used to estimate harmonic parameters of unknown signals.

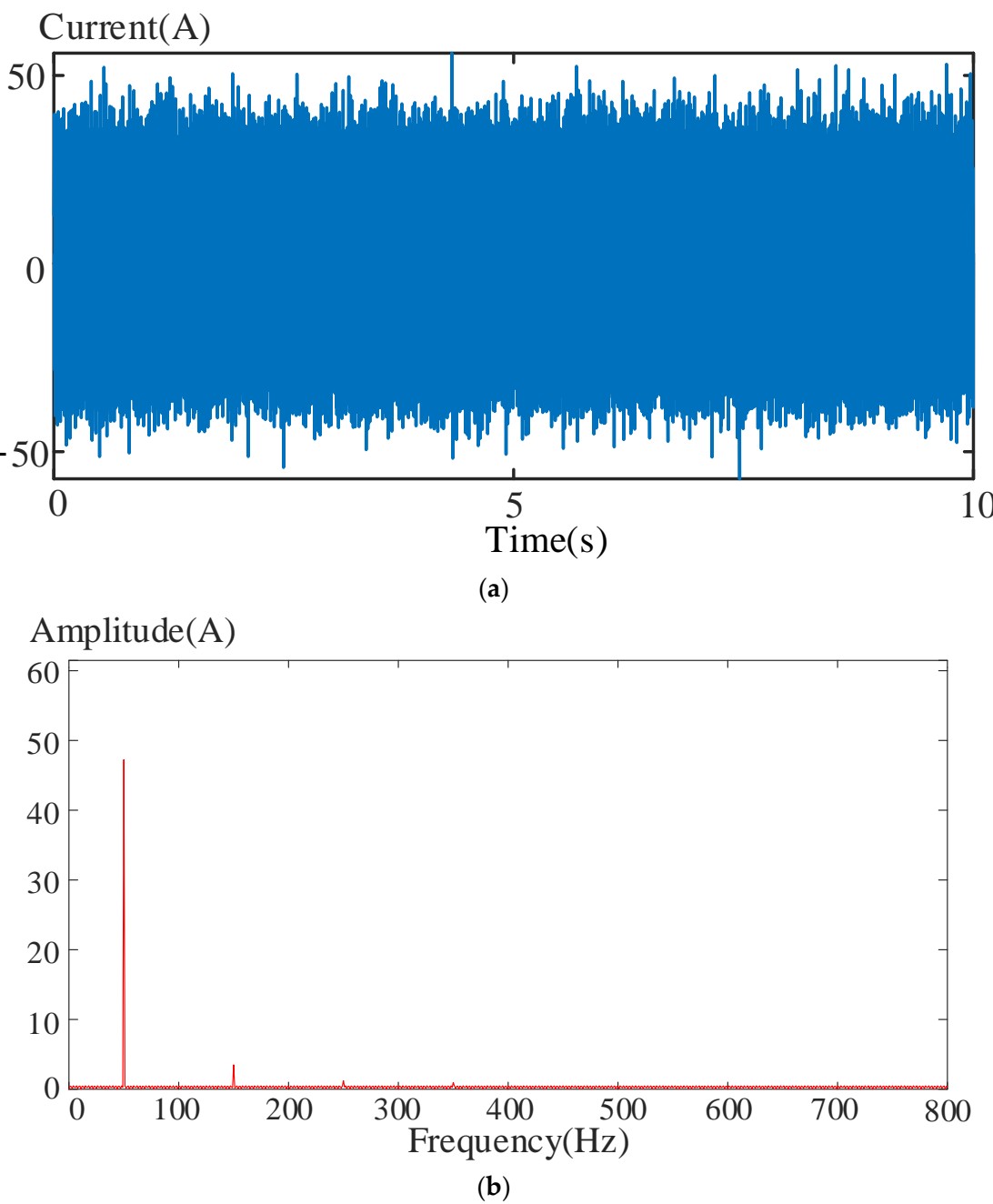

**Figure 5.** (**a**) Current recorded data. (**b**) spectrum.

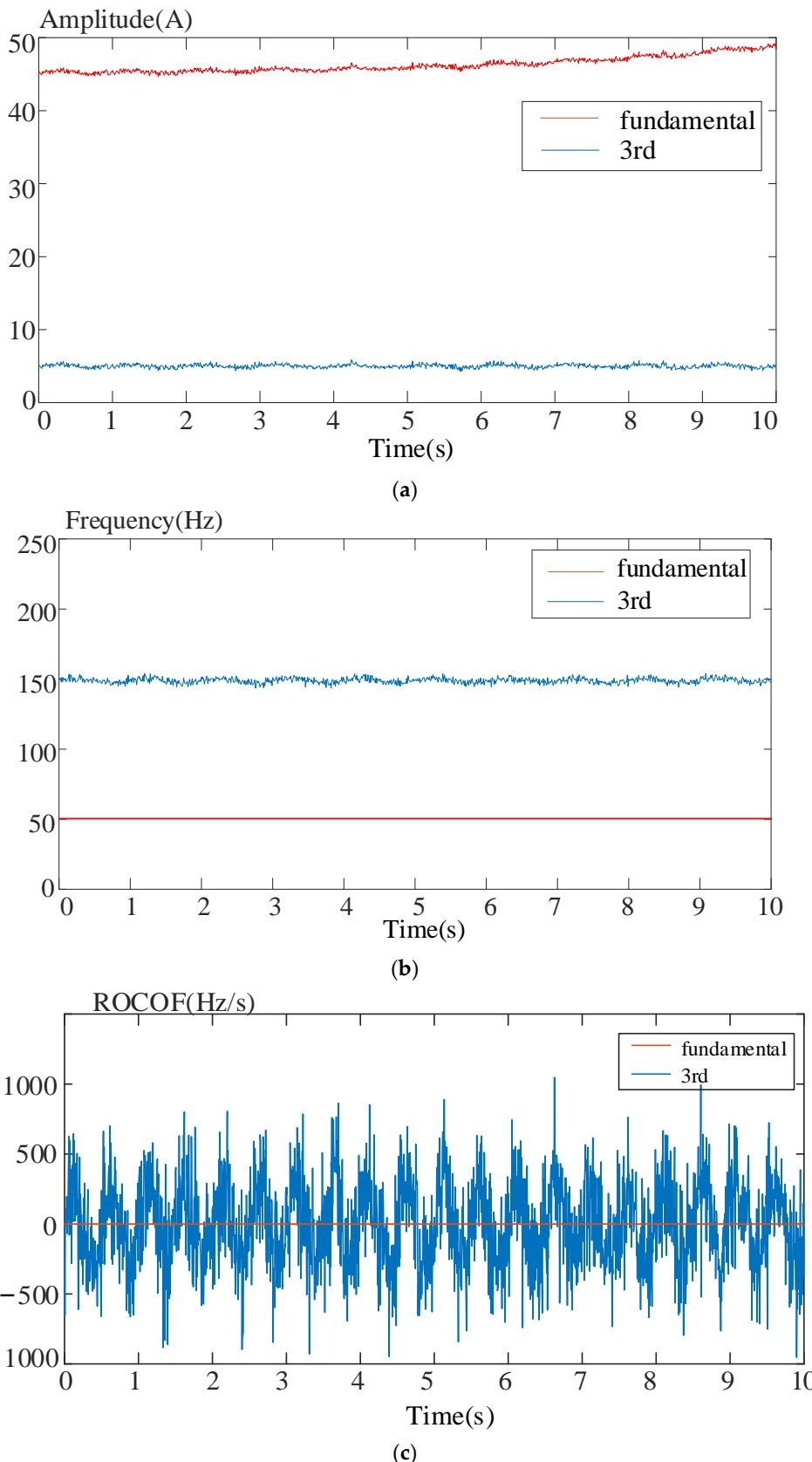

**Figure 6.** Parameter estimates of the BMP. (**a**) Amplitude estimates. (**b**) Frequency estimates. (**c**) ROCOF estimates.

## 5. Conclusions

In this paper, a new wideband dynamic phase measurement algorithm is proposed that offers efficient processing of wideband dynamic phase s containing DDC components. The algorithm is based on the idea of regularized sparse capture matrix. The results of phase estimation are obtained by solving the sparse regularization problem. The results obtained from simulation and experiment based studies show that the proposed algorithm identifies the key information of the wideband dynamic phase and significantly reduces the computational complexity. This demonstrates the ability to obtain more accurate results in a short time and thus effectively detect the transient characteristics of broadband signals. Meanwhile, the measurement results can meet the test requirements of M-class PMU under static and dynamic conditions such as noise interference, interharmonic interference, and frequency ramp. However, the algorithm is still not able to solve the dense frequency signal analysis problem effectively. In order to continuously improve the accuracy of phase estimation, the next research direction is to consider the algorithm of broadband phase measurement under dense frequency signals.

**Author Contributions:** Conceptualization, Y.G.; methodology, Y.G.; software, A.C.; validation, A.C.; investigation, H.X.; resources, H.X.; data curation, H.X. and A.C.; writing—original draft preparation, Y.G.; writing—review and editing, Y.G.; supervision, H.X.; project administration, H.X.; funding acquisition, Y.G. All authors have read and agreed to the published version of the manuscript.

**Funding:** This study was supported by the Key R&D Project of Sichuan Science and Technology Department, Sponsor: Yingwei Zhu, Project Number: 2021YFG0351.

**Data Availability Statement:** The data presented in this study are available on request from corresponding authors.

**Conflicts of Interest:** The authors declare no conflict of interest.

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
