# Peer review of "Broadband Dynamic Phasor Measurement Method for Harmonic Detection"

_electronics, doi:10.3390/electronics11111667_

Round 1

Reviewer 1 Report

In this paper, broadband dynamic signals with inter-harmonics and decaying DC (DDC) components, generated by nonlinear loads and distributed energy sources connected to the power system, cause power quality and measurement problems. So, proper monitoring and analysis of harmonics and interharmonic phasors require good phasor estimate methods. For this, a two-part algorithm is proposed by the authors. First, they used least squares to get precise DDC components. Then, two-part model of broadband dynamic harmonic phasor is established. Harmonic vector estimation solves the sparse acquisition regularization problem. SBI is used to estimate the harmonic phasor measurement and reconstruct the original signal. Through simulation and performance testing, their proposed approach increases phasor measurement and estimate accuracy and provides a reliable theoretical basis for PMU measurement.

This study provided simulation results that conform to their theoretical foundation of their proposed approach. The step-by-step procedure on how the simulation was made using the proposed method is not clear. Although performance tests were performed, it is better that the authors present measurement validations and analysis to suffice their proposed technique. Moreover, it is preferred that measurement tools, equipment, and setup be reflected on the paper so that the readers will understand how the study was conducted.

Reviewer 2 Report

This paper first uses the Sinc interpolation function to model the attenuated DC component parametrically and obtain the accurate DDC component by the least-squares method. the comments are below:

  • Please highlight the research gap.
  • what is the novelty of this work if compared to available works in literature?

Reviewer 3 Report

A work devoted to the measurement of harmonic composition is proposed. The authors offer an overview of this issue and work has been done using the proposed algorithms. I think that the article can be published in the current version.

Author Response

Thank you very much for reviewing my manuscript.

Round 2

Reviewer 1 Report

The authors have already met and satisfied the comments and suggestions made.

Reviewer 2 Report

accept